# Universal model of individual and population mobility on diverse spatial scales

Xiao-Yong Yan[1], Wen-Xu Wang [2], Zi-You Gao[1] & Ying-Cheng Lai[3]

Studies of human mobility in the past decade revealed a number of general scaling laws. However, to reproduce the scaling behaviors quantitatively at both the individual and population levels simultaneously remains to be an outstanding problem. Moreover, recent evidence suggests that spatial scales have a significant effect on human mobility, raising the need for formulating a universal model suited for human mobility at different levels and spatial scales. Here we develop a general model by combining memory effect and population-induced competition to enable accurate prediction of human mobility based on population distribution only. A variety of individual and collective mobility patterns such as scaling behaviors and trajectory motifs are accurately predicted for different countries and cities of diverse spatial scales. Our model establishes a universal underlying mechanism capable of explaining a variety of human mobility behaviors, and has significant applications for understanding many dynamical processes associated with human mobility.

[1] Institute of Transportation System Science and Engineering, Beijing Jiaotong University, Beijing 100044, China. [2] School of Systems Science and Center for Complexity Research, Beijing Normal University, Beijing 100875, China. [3] School of Electrical, Computer and Energy Engineering, Arizona State University, Tempe, AZ 85287, USA. Correspondence and requests for materials should be addressed to W.-X.W. (email: wenxuwang@bnu.edu.cn) or to Z.-Y.G. (email: zygao@bjtu.edu.cn) or to Y.-C.L. (email: Ying-Cheng.Lai@asu.edu)

Human movements typically occur in spatial regions/domains, such as countries or cities, which can have vastly different scales. For example, in China or the United States, the size of the region can be on the order of millions of square kilometers, while in small countries such as Belgium, the domain size is only about 10 km. (Here we regard international travel as atypical and exclude it from our consideration.) Comparing Belgium with China or the United States, the difference in the spatial scale in terms of areas is at least two orders of magnitude. Typical examples of human movements at both the individual and population levels in countries of diverse size are shown in Fig. 1. Overall, for human mobility, there are large scales exemplified by countries such as China and the United States, and small scales as represented by small countries or big cities in a large country.

A remarkable discovery in complexity science in the past decade is that human mobility obeys certain universal scaling laws[1–20]. It was recently revealed[16], however, that human mobility at small spatial scales tends to exhibit different scaling behaviors. Existing models of human mobility are tailored at understanding and characterizing scaling laws either at large (e.g., big country) or small (e.g., city) scales—we lack a universal model capable of quantifying human movements across all spatial scales. Another deficiency of existing models is that they can explain human mobility patterns either at the individual or at the population level, but not both. The purpose of this paper is to develop a model to fill this gap in our knowledge about human mobility.

To understand the dynamics of human movements and to uncover scaling laws underlying human mobility are of fundamental importance as they are relevant to problems such as disease control, social stability, congestion alleviation, information propagation, and e-commerce[21–28]. Data-based modeling research on human mobility started about a decade ago[1], where the trajectories of bank notes were traced over a reasonably long time period. On the basis of empirical data, the study unveiled two scaling laws on geographical scales: (1) the distribution of the traveling distance exhibits a power-law decay, which can be described by Lévy flights, and (2) the probability of remaining in a small region exhibits an algebraically long tail with an exponential cutoff, which is characteristic of a superdiffusive behavior. Existence of universal patterns in the statistical description of human movements was hinted at through an analysis of the mobile phone data[6], and the issue of predictability of human movement patterns was also addressed[16,29]. The correlation between human movements in the cyberspace and in the physical space has also been studied through big data analysis[15,20]. Quite recently, a scaling law connecting human mobility and social interaction (communication) patterns was uncovered[30].

From a modeling perspective, the classic gravity model[31] represented perhaps the earliest attempt to mathematically understand the mobility flow between two locations. For human mobility on large spatial scales, e.g., as revealed by the trajectories of bank notes, a two-parameter continuous-time random walk model was derived to explain, at a detailed and quantitative level, the empirically observed scaling laws[1]. A statistical, self-consistent microscopic mobility model[2] and a macroscopic model, the so-called radiation model[3–5] that takes into account local mobility decisions, were developed. Inspired by these models, a variety of alternative mechanisms aiming at understanding and characterizing the empirical scaling laws obtained from data have been conceived[6–20]. Most classic gravity-based

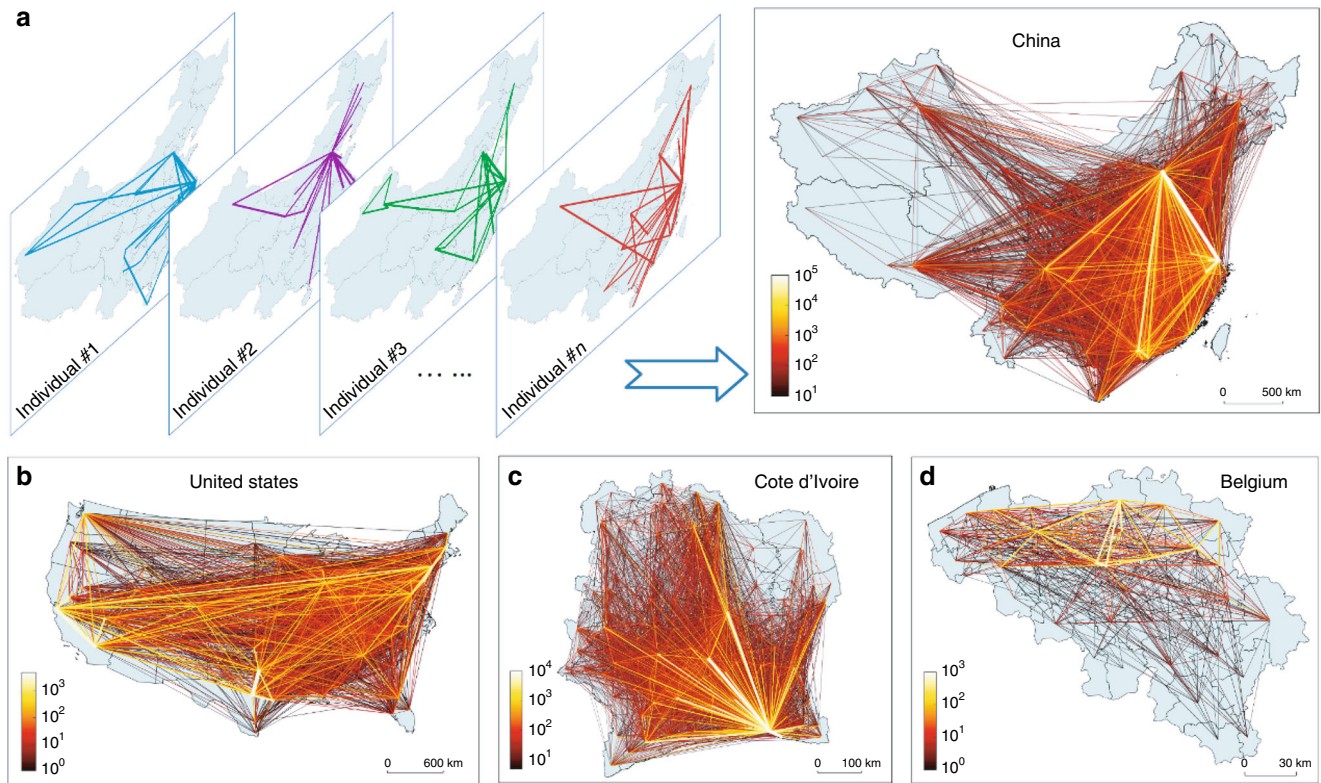

**Fig. 1** Real-world examples of individual trajectories and collective movements. **a** Four examples of an individual trajectory from an empirical data set from mainland China and the corresponding collective movements. **b–d** Collective movements embedded in the data sets from the continental United States, Cote d'Ivoire, and Belgium. Here the color bar represents the amount of mobility flux among locations per unit time, where a brighter (darker) line indicates a stronger (weaker) flux. Note that the spatial scales associated with these data sets are drastically different

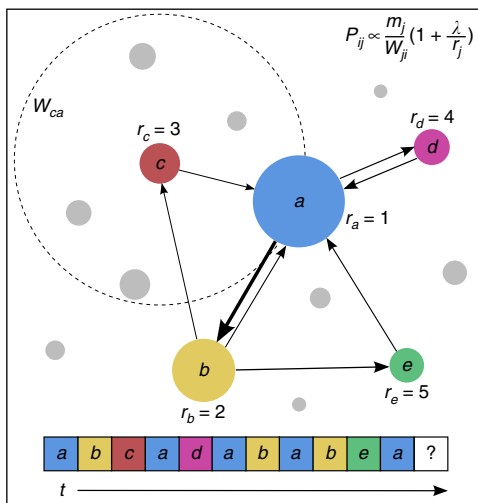

$$P_{ij} \propto \frac{m_j}{W_{ji}}\left(1 + \frac{\lambda}{r_j}\right)$$

**Fig. 2** Model illustration. A typical trajectory visiting the five locations denoted by letters $a - e$ with different colors is indicated at the bottom: $a \rightarrow b \rightarrow c \rightarrow a \rightarrow \dots$. The size of the circles that contain a letter indicates the relative attractiveness of the corresponding location as characterized by the index $r_j$, where $r_1$ is the most attractive location, $r_2$ is the second most attractive one, and so on. The dashed circle centered at location $c$ indicates that the travelers moves from $c$ to $a$, whose radius is the distance between $c$ and $a$, and the total population within the dashed circle is $W_{ca}$. The model contains a single parameter, $\lambda$, which can be determined from each set of empirical data

macroscopic models are static in the sense that they assume certain mobility decisions so that the transition matrix between spatial locations can be constructed through the corresponding distances and population distribution. While the models can explain the scaling laws to a certain degree, the detailed dynamical features associated with human movements at the individual level are often lost, which are important to understand issues such as the spreading speed and range of diseases. The existing microscopic models[2,8,15,20,32–34] can capture the individual movements and the associated scaling laws, but the mutual interactions among the individuals at the population level were largely ignored.

In this paper, we aim to articulate a human mobility model capable of predicting statistical and scaling behaviors on all spatial scales at both the individual and population levels. To accomplish this goal, it is necessary to identify the general mechanisms underlying human movements independent of the spatial scale. An essential element upon which any human mobility modeling is built is the transition probability for an individual to move from one location to another at any time, from which all kinds of scaling laws can be derived. There are two basic elements that we exploit to construct the transition probability. First, locations are differentiated according to their relative attractiveness, so human movements tend to be biased toward the more attractive locations. This can be modeled by assigning each location a fixed amount of attractiveness. The second element is the memory effect, by which individuals tend to move preferentially to previously visited locations. At a quantitative level, the memory effect can be taken into account by assigning previously visited locations with a relatively high amount of attractiveness. As a result, in our model a location contains two kinds of attractiveness: one simply determined by the population at the location, which is analogous to that in the radiation model[3] or the population-weighted opportunities model[16], and another determined by the memory effect. For any location, the transition probability for an individual to move into

it is proportional to its total attractiveness. More specifically, before making a movement, an individual evaluates the attractiveness of all the available destinations and then moves to a specific destination according to the transition probability. With the transition probability so determined, our model contains a single parameter, which can be determined from each set of empirical data. After the parameter is fixed, our model can simultaneously generate a number of key scaling laws at both the individual and population level, as well as the trajectory motifs, which are in good agreement with empirical results from data associated with arbitrarily spatial scales.

## Results

**Model**. Typical examples of trajectories of human movements at the individual and population levels are shown in Fig. 1. Our aim is to develop a model that can capture the statistical features and predict the scaling laws associated with trajectories at both the individual and population levels, regardless of the spatial scale. A key quantity is the transition probability. In the recent population-weighted opportunities model[16], the transition probability to a destination is proportional to its attractiveness. This probabilistic rule determines the movement of any individual but for only one time step. In order to quantify the statistical behaviors and the scaling laws, sufficiently long trajectories of a large number of individuals are needed.

An important characteristic of human movement, which distinguishes its dynamics from the diffusion dynamics of physical particles, is the memory effect[2,20,33]. In particular, individuals tend to frequently return to previously visited locations. There are different approaches to taking memory effect into account. For example, in the exploration and preferential return (EPR) model[2], it was assumed that the probability for an individual to visit a new location is $p \propto S^{-\gamma}$, where $S$ is the total number of locations that the individual has already visited and $\gamma > 0$ is a model parameter. The probability for the individual to visit a previous location is thus $1 - p$. The algebraic dependence of $p$ on $S$ indicates that the more locations that an individual has visited, the smaller the probability would be for him/her to explore any new location. That is, there is a strong preference for an individual to move among locations that have been visited previously. The model also assumes[2] that the probability for an individual to move to a previously visited location is proportional to the frequency at which it has been visited. This model can successfully reproduce the visiting frequency distribution of the locations obtained from empirical data, as well as the rate of increase in the number of locations. A subsequent model[33] emphasizing the importance of the memory effect assumes that the probability distribution of the return time interval, $P(\tau)$, is known. An individual chooses a value of $\tau$ from the distribution to determine the location that he/she wishes to return to. While this model can reproduce the empirically obtained rate of increase of new locations, the choice of $P(\tau)$ is mostly heuristic.

The basic idea underlying the development of our model is that the attractiveness of a location for an individual is determined by both the memory of the individual and the population at the location. Let $A$ be a quantity measuring the effect of memory on the attractiveness of a location to an individual. It is reasonable to assume that a more attractive location can in general impose greater impression on the visitors, resulting in stronger memory and, consequently, enhancing the probability for the individual to visit the location in the future. That is, the attractiveness of a location will be reinforced by good memory and vice versa.

To characterize $A$ in a quantitative manner, we rely on empirical evidence of human travel, in which the frequencies of

**Table 1 Description of empirical data sets**

| Country | Data type | Number of cities | Population | Total traveling steps | λ | GDP per capita ($) |
|---|---|---|---|---|---|---|
| China (mainland) | Sina Weibo check-ins | 340 | 1,571,056 | 4,976,255 | 35 | 8141 |
| US (lower 48 states) | Foursquare check-ins | 125 | 32,040 | 194,730 | 32 | 56,084 |
| Cote d'Ivoire | Mobile phone CDR | 237 | 229,335 | 8,747,801 | 50 | 1325 |
| Belgium | Gowalla check-ins | 43 | 1352 | 21,156 | 25 | 40,529 |

Information of data sets DS1–DS4 and the values of their memory parameter λ are shown
The four data sets correspond to four countries with different spatial scales, and the features of the four countries from the data sets include the number of cities, population, total traveling steps, and GDP per capita. Here, the GDP data is obtained from http://www.imf.org. An individual traveling from one city to another represents one travel step. The number of total traveling steps is the sum of all recorded individual steps

individual visit to different locations are distributed according to the Zipf's law[6]. It is thus reasonable to assume that $A$ is distributed in a similar manner. That is, the Zipf's law stipulates that visitors rank the locations visited such that the probability to visit a location is inversely proportional to its rank. For example, the probability of visiting the most frequently visited location is $A_1 = \lambda/1$ and the probability of visiting the second most frequently visited location is $A_2 = \lambda/2$, and so on, where $\lambda$ is a constant. Due to an aging effect, the most frequently visited location is often the "oldest" one. We can thus replace the rank of a location by the order with which it is visited. These considerations lead to the following formula to quantify the memory effect:

$$A_j = 1 + \frac{\lambda}{r_j}, \qquad (1)$$

where $\lambda$ is a parameter characterizing the strength of the memory effect and the index $r_j$ denotes that location $j$ is the $r$th newly visited location associated with the movement trajectory. The unity in the formula represents the initial attractiveness of location $j$ that has not been visited, i.e., if a location has not been visited, its $A$ value is always unity.

Following the classic gravity model, we assume that the attractiveness of a location is proportional to its population. Let $B$ be a quantity characterizing the population-induced effect on the attractiveness of a location, and let $m_i$ be the population of location $i$ and $N$ be the total number of locations that can possibly be toured by all the individuals. As illustrated in Fig. 2, the attractiveness $B_{ij}$ for a visitor to travel from location $i$ to destination $j$ is

$$B_{ij} = \frac{m_j}{W_{ji}}, \qquad (2)$$

where $W_{ji}$ is the total population in the circular region centered at $j$, the radius of which is the distance between locations $j$ and $i$. Note that $B_{ij}$ reflects the competition for opportunities among different locations. For instance, if a traveler at location $i$ wishes to visit a potential destination $j$, more populations between $i$ and $j$ imply more fierce competitions for limited opportunities at those locations, leading to a lower probability of being offered some opportunity. It is thus reasonable to assume that the attractiveness $B_{ij}$ of destination $j$ for a visitor from location $i$ is inversely proportional to the population between $i$ and $j$, as quantified by formula (2).

The transition probability $p_{ij}$ of traveling from location $i$ to $j$ is then proportional to both $A_j$ and $B_{ij}$, which can be written as

$$p_{ij} \propto \frac{m_j}{W_{ji}}\left(1 + \frac{\lambda}{r_j}\right). \qquad (3)$$

We see that the model contains a single adjustable parameter, the memory strength $\lambda$, that can be determined from empirical data. For any location $i$, we place a number of travelers. Each traveler is

assigned a number $L$, the total number of movement steps, which can be obtained from an actual distribution from empirical data. A traveler thus executes a trajectory of length $L$ and, at each step, he/she moves to a destination according to the transition probability $p_{ij}$.

**Model prediction and validation**. Our model, as illustrated in Fig. 2, is capable of predicting the statistical behaviors of human mobility at both the individual and population levels, regardless of the spatial scale. At the individual level, we focus on the following quantities: (I1) the total number of locations visited by time $t$, (I2) return time distribution to any location, (I3) distribution of frequency of visits to a location, and (I4) emergence of traveling motifs and their probability of occurrence in a long trajectory. At the population level, we seek to predict: (P1) distribution of the travel distance of collective movement and (P2) distribution of the number of traveling steps between two locations.

To validate the model predictions, we employ four empirical data sets, as illustrated in Fig. 1. They are: (DS1) record of user check-ins at Sina Weibo in mainland China (Fig. 1a), (DS2) check-in record of the site Foursquare[35] for users in the continental United States (Fig. 1b), (DS3) communication record of mobile phone users in the whole country of Cote d'Ivoire[36] (Fig. 1c), and (DS4) check-in record of the site Gowalla[37] in Belgium (Fig. 1d). Each data set contains spatial and temporal information about continuous user mobility, from which data of movements among various locations (e.g., cities) can be extracted (Methods). The single free parameter $\lambda$ can be determined from data (Methods). We obtain $\lambda = 35, 32, 50$, and 25 for data sets DS1–DS4, respectively. A heuristic observation is that $\lambda$ assumes a relatively smaller value for a better developed country (Methods). An explanation is that, in general, in a country with a higher gross domestic product (GDP), individuals can afford more travel, leading to more visited locations and a higher probability of exploring new places. In contrast, in a country with a lower GDP, it is more difficult for people to travel frequently and they tend to stay in their home cities and familiar places. That is, a higher GDP induces a weaker memory effect and a higher probability of visiting new locations, as reflected by the smaller values of the memory strength $\lambda$ in well developed countries.

Since the data sets contain no information about the individuals' cities of residence, for each individual, we assign the city that he/she signs in with the highest frequency as his/her home city and use it as the initial location in the model. From the data, we calculate the distribution $P(L)$ of the total number of times of movement and choose $L$ accordingly, which is effectively the trajectory length for each individual. It is worth noting that an effective way to test our mobility model is to use the same distribution of the trajectory length as that from the empirical data. We also study analytically the impact of trajectory length on the statistical properties of mobility at both the population and individual levels, with the finding that, for a sufficient number of moving steps, simulation results are in good agreement with the

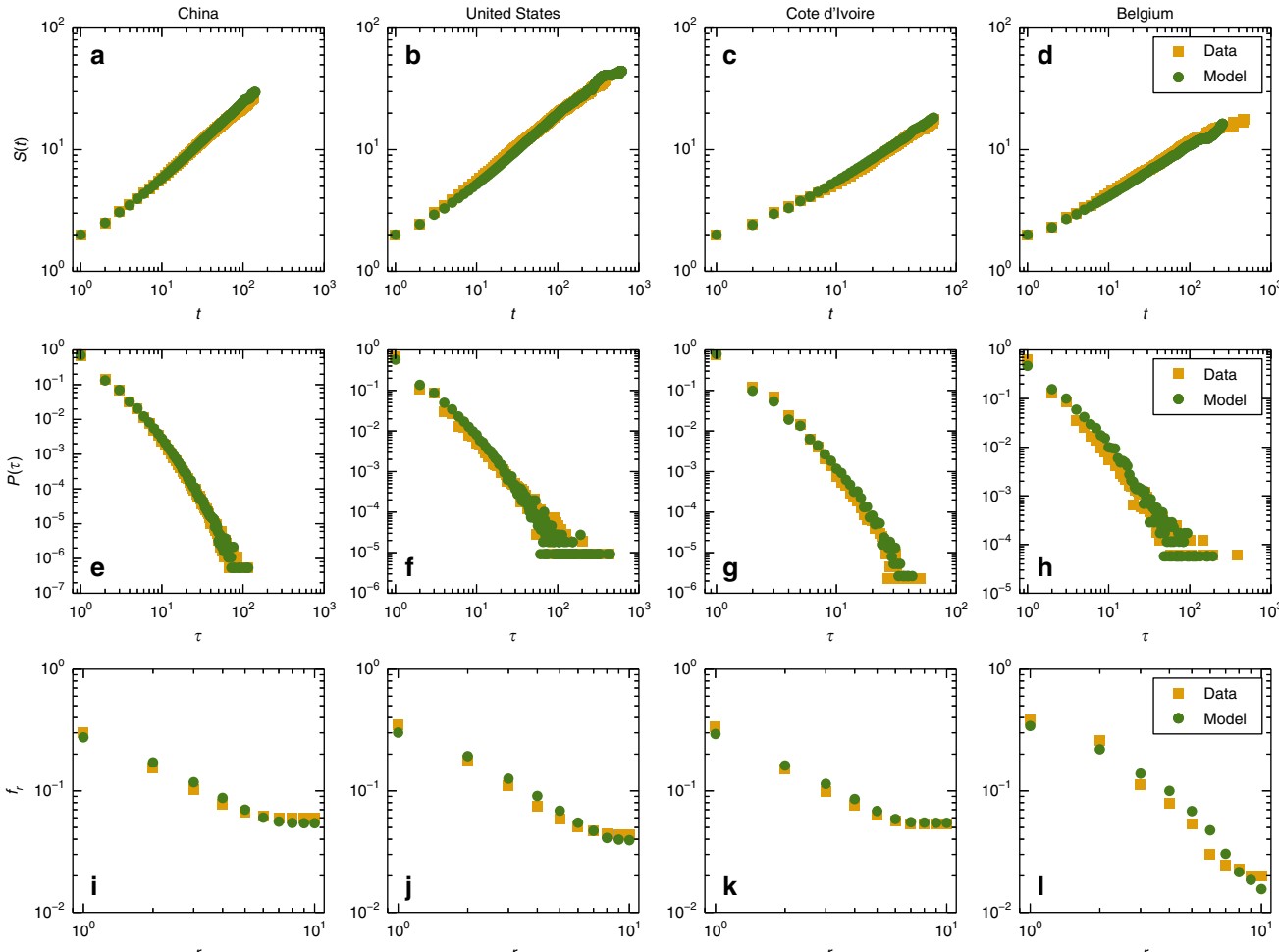

**Fig. 3** Model-predicted and empirical statistical behaviors of human mobility at the level of individual trajectories. **a–d** Algebraic increase with time $t$ in the total number of locations visited within $t$, **e–h** algebraic decaying behavior in the return time distribution, and **i–l** Zipf-like frequency distribution of visits. In all panels, the green color specifies results predicted by our model, and orange denotes the empirical results obtained directly from data. There is a generally strong agreement between the two types of results. The algebraically increasing and decreasing behaviors in **a–d** and **e–h**, respectively, are manifestations of the memory effect

analytical prediction. This indicates that the trajectory length has little effect on the mobility patterns in the long time regime. For the empirical data (Table 1), the total numbers of steps are much larger than those on the population records, so the mobility patterns produced by our model are stable and robust.

Figure 3a–d show, for the data sets DS1–DS4, respectively, the model-predicted algebraic increase with $t$ in the total number of locations visited by time $t$ (green), together with the corresponding results calculated directly from the data (orange). We obtain an excellent agreement between model prediction and the empirical result. The algebraically increasing behavior, as opposed to an exponential growth in the number of cities visited in certain time, is a natural consequence of the memory effect, which is a key ingredient in our model. Figure 3e–h show the predicted and actual return time distributions for the data sets DS1–DS4, respectively, which are algebraic. There is again an excellent agreement between the model prediction and the empirical result. The algebraic decay in the return time distribution can also be attributed to the memory effect. Thus, both the algebraically increasing behavior in Fig. 3a–d and the algebraically decaying behavior in Fig. 3e–h are manifestations of the same memory effect. Figure 3i–l show, for the data sets DS1–DS4, respectively, the model-predicted and empirical frequency distributions of

visits to all locations, which agree with each other reasonably well and follow approximately the Zipf's law. The emergence of the Zipf-like scaling behavior is indicative of the heterogeneity in the location attractiveness, an assumption of our model. The results in Fig. 3a–l validate our model with respect to the statistical behaviors of individual trajectories.

A characteristic of human mobility is the emergence of motifs associated with movement trajectories[38], which are referred to as certain simple and fixed patterns of visit that occur repeatedly in a long trajectory. For an individual initially at his/her home location (the one visited with the highest frequency), a motif is defined as a successive sequence of locations visited with the last location being the initial one. From the empirical data, we identify nine distinct motifs (shown at the top in Fig. 4) and calculate the frequencies of their occurrences from the entire data set. With parameter $\lambda$ extracted from the data, our model can generate long trajectories from which the possible motifs and their frequencies of occurrence can be determined. Remarkably, our model yields exactly the same set of motifs with frequencies that agree with the empirical results reasonably well, as shown in Fig. 4. Due to the significance of travel motifs in determining the microscopic mobile patterns of travelers, the agreement provides further validation of our model at the individual level.

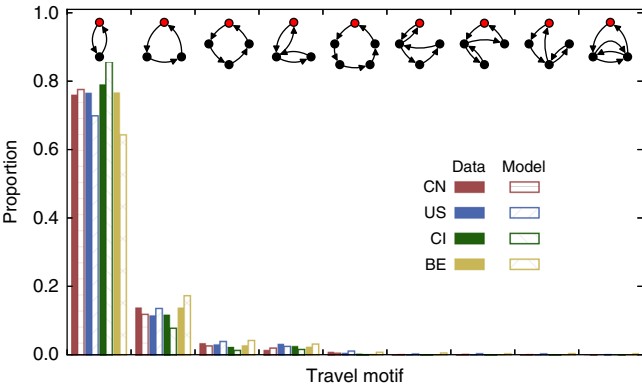

**Fig. 4** Frequency of occurrence of motifs associated with movement trajectories. From the four empirical data sets, nine distinct motifs contained in movement trajectories are identified. Exactly the same motifs are obtained from model generated trajectories. The frequencies of occurrence of the motifs from the model and from actual trajectories agree with each other reasonably well. There exist many more types of possible motifs. However, the nine types included here have the highest frequencies of occurrence, the sum of which exceeds 0.97. The frequency of the 10th highest motif, for example, is <0.001. Considering the nine motifs thus suffices

Our model also has strong predictive power for human movements at the population level on all spatial scales. As shown in Fig. 5, the predicted behaviors of $P(d)$ and $P(T)$, the distributions of the travel distance and of the number of traveling steps between two locations, agree well with the statistical results from the empirical data. In particular, Fig. 5a–d show, for the data sets DS1–DS4, respectively, that $P(d)$ decays exponentially. Figure 5e–h reveal that $P(T)$ exhibits a robust algebraic scaling for all four data sets. Figure 5i–l demonstrate that the model-predicted and real values of $T$ are nearly statistically indistinguishable (albeit with fluctuations). Our model is then universally applicable to characterizing human movements across vast different spatial scales at the population level.

**Theoretical analysis**. In our model, the fixed amount of attractiveness of a location is calculated based on its population. Since the population distribution is typically highly heterogeneous without a closed mathematical form, it is not feasible to treat our model exactly. To gain analytic insights, we consider a simplified model obtained by imposing the approximation that the population is uniformly distributed among the locations, and focus on analytically predicting the individual trajectories and the collective mobility pattern with a special emphasis on the role of the memory effect. Although the simplified model deviates from real scenarios, the analytical predictions enable a good understanding of the real mobility patterns at both the individual and population levels.

In the simplified model, an individual travels among $N$ locations. At each time step, the probability to move to a destination is proportional to its attractiveness. At $t = 0$, the initial attractiveness is identical (unity) for all locations. During the travel, the attractiveness of the $r$th first visited location is updated to $1 + \lambda N/r$, where $\lambda > 0$ is a parameter. The model describes essentially a random-walk process with time varying location attractiveness, with parameter $\lambda$ characterizing the memory strength of (or preference to) locations previously visited. For $\lambda = 0$, the model is reduced to an unbiased random walk. For $\lambda = \infty$, the walker can travel between only the first two locations. The total number $S(t)$ of locations visited by time $t$ can be used to characterize how fast the underlying mobile process takes place.

For a uniform random walk, $S(t)$ increases with $t$ linearly: $S(t) \propto t$. For the EPR model[2], $S(t)$ increases with $t$ but in a sublinear fashion: $S(t) \propto t^{\beta}$ with $0 < \beta < 1$. For our random walk model with memory, $S(t)$ can be obtained analytically (Supplementary Note 1), as shown in Fig. 6a. We see that, as the memory strength parameter $\lambda$ is increased, the overall rate of increase in $S(t)$ becomes smaller. In addition, for a fixed value of $\lambda$, the time derivative of $S(t)$ tends to increase with time, which is consistent with the result from real data (c.f., Figs. 1a and 3 in ref. [2]).

Another characteristic quantity is $f_r$, the distribution of the frequency of visit to location $r$. For an unbiased random walk, $f_r$ is uniformly distributed. For the EPR model[2], $f_r$ decays algebraically: $f_r \propto r^{-\alpha}$, where $\alpha > 0$ is a constant. For our model, we analytically obtain (Supplementary Note 1)

$$f_r \propto \frac{\lambda S}{r} + 1 - \lambda. \tag{4}$$

For $\lambda = 0$, Eq. (4) reduces to a uniform distribution. For $\lambda = 1$, we recover the Zipf's law for $f_r$. Figure 6b shows the analytic and simulation results of $f_r$ for a number of $\lambda$ values, where the curves represent the theoretical prediction. We see that, for $\lambda > 1$, there is an apparent deviation from the Zipf's law, as signified by the emergence of an exponential cutoff toward the tail end of $f_r$. The physical meaning is that, as the memory effect is intensified, a walker tends to travel among only a few locations.

The return time distribution $P(\tau)$ is defined as the probability for a walker to return to one of the previously visited locations after $\tau$ steps, which is also a reflection of the memory effect. In our model, $P(\tau)$ contains two different algebraic terms but with the same exponent $-1$ (Supplementary Note 1). As $\lambda$ is increased, $P(\tau)$ tends to a single algebraic distribution with essentially zero values near the tail, indicating an extremely low probability for the walker to return to a previous location after many time steps. Figure 6c shows the analytic and simulation results of $P(\tau)$ from our model. The agreement is reasonable, and the deviation of the analytic from the simulation result in the large $\tau$ region is due to the finite time used in the simulation.

Finally, we remark on an appealing feature of our model. Consider the probability for the walker to choose a new location at the next time step. Analysis of our model leads to (Supplementary Note 1)

$$P_{\text{new}} = \frac{1}{1 + \lambda(\ln S + C)}. \tag{5}$$

Thus, in our model, $P_{\text{new}}$ decreases with $S$, which occurs naturally as a consequence of a basic and intuitive assumption, namely the memory effect.

A further simplification of the model by assuming that each individual can move one step only renders analytically predictable collective mobility patterns at the population level. In particular, we place $m$ individuals at each location and exploit the previously discovered[39], common fractal feature in the spatial distribution of locations in the real world: $W_{ji} \propto d_{ij}^{D}$, where $D$ is the fractal dimension. Equation (3) can be formulated as (Supplementary Note 2)

$$T_{ij} = m_i p_{ij} \propto \frac{m_i m_j}{d_{ij}^{D}}, \tag{6}$$

where $T_{ij}$ is the total number of traveling steps from $i$ to $j$ for the whole population. Equation (6) is a standard gravity model with a power-law distance function. Since Eq. (6) indicates that the number of traveling steps $T$ between two locations of distance $d$ is $T(d) \propto d^{-D}$, the travel distance distribution is given by the same form: $P(d) \propto d^{-D}$, as validated by Fig. 6d for four typical fractal domains. The number of location pairs with distance $\leq d$ in a

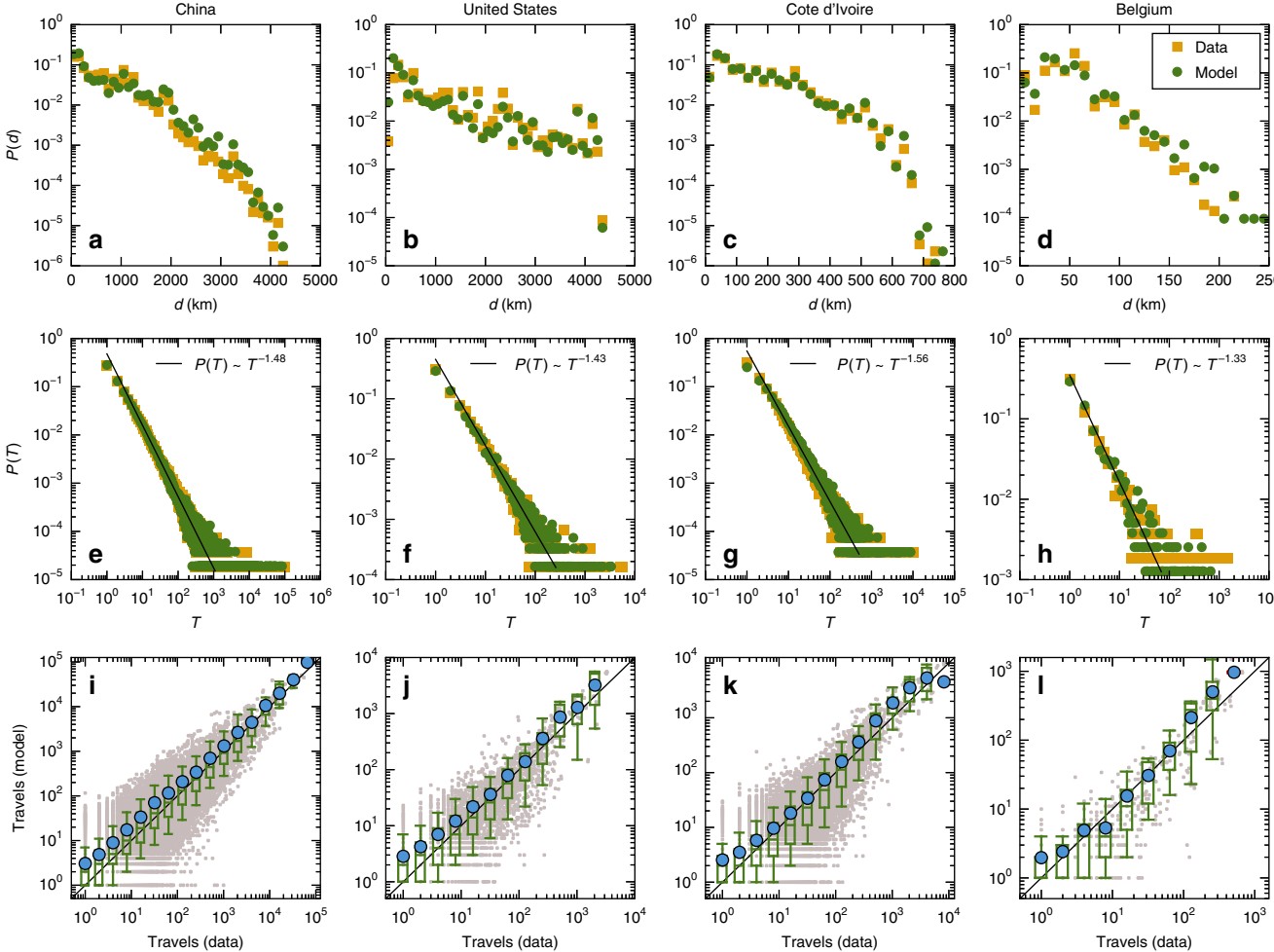

**Fig. 5** Model prediction at the population level and validation. For the four empirical data sets with vast difference in the spatial scale, model-predicted and real distributions of travel distance $d$ (**a**–**d**) and the number of traveling steps, $T$, between two locations **e**–**h**. The lines in **e**–**h** represent a power-law fit to the real data. **i**–**l** Statistical display of the model-predicted and real values of $T$ for the four data sets, respectively, which are nearly indistinguishable. The gray points are scatter plot for each pair of locations. The blue points represent the average number of predicted travels in different bins. The standard boxplots represent the distribution of the number of predicted travels in different bins of the number of observed travels. A box is marked in green if the line $y = x$ lies between 10% and 91% in that bin and in red otherwise

fractal domain is $N(d) \propto d^D$. Thus, the number of traveling steps $T$ obeys a power-law distribution (Supplementary Note 2):

$$P(T) \propto T^{-2}. \tag{7}$$

It is worth noting that the algebraic exponent −2 is universal, regardless of the fractal dimension $D$ of the location distribution in the simplified model, as shown in the insert of Fig. 6d. However, in the real world, the heterogeneous nature of the population distribution at different locations can cause a deviation of the exponent from −2. As shown in Fig. 5e–h, the fit of the empirical data demonstrates that their algebraic exponents range from −1.33 to −1.56 (see Supplementary Note 2 for a detailed explanation of the effect of heterogeneous population distribution on the algebraic exponent). Nonetheless, the power-law distribution predicted by our simplified model is robust, which captures the essential features of the collective mobility patterns in the real world.

In the development of our human mobility model, Zipf's law is naturally included as an essential component. However, the Zipf's law is closely related to diminishing exploration. To elucidate the interplay between the two, we articulate an extended individual mobility model based on the generalized Zipf's law. This is guided

by the previous evidence that there are situations where individuals tend to choose locations to travel into by following the generalized Zipf's law[40]. Specifically, we assume $f \propto r^{-\zeta}$, where the exponent $\zeta > 1$ is an adjustable parameter. For the extended model, we analytically obtain $P_{\text{new}} = \rho S^{-\gamma}$, where $\gamma = \zeta - 1$. We see that the formula of $P_{\text{new}}$ is a direct manifestation of the basic assumptions in the EPR model[2]. This suggests that the generalized Zipf's law and the power-law relation between $P_{\text{new}}$ and $S$ have a mutually causal relationship, and the individual mobility models based on the former and latter are equivalent to each other. For the extended model, we also derive the return time distribution $P(\tau)$ for sufficiently large values of $S$. A detailed description of the extended model, the analysis, and results are presented in Supplementary Note 1.

## Discussion

The past decade has witnessed a great deal of efforts into uncovering and understanding the general dynamical behaviors of human mobility. A variety of real data sets have been analyzed, leading to a spectrum of mathematical models being devised to explain the phenomena revealed by data. While universal scaling laws have been unveiled, it turns out that spatial scale has a significant effect on the dynamics. In particular, human

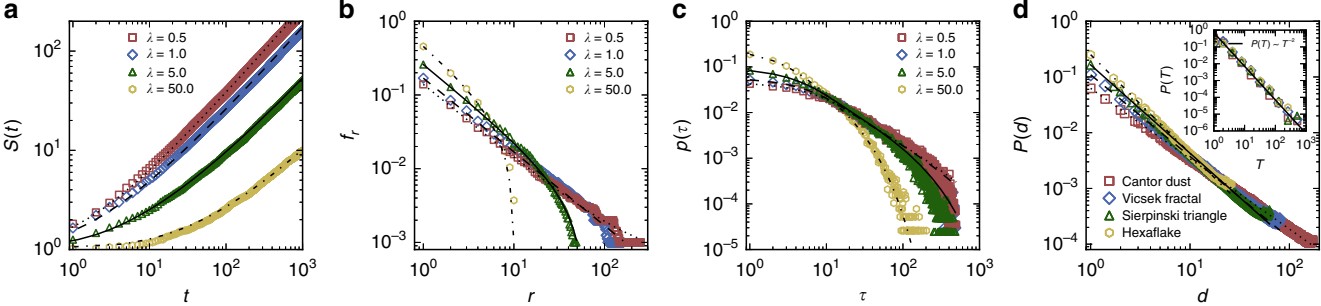

**Fig. 6** Analytic predictions of a simplified model in comparison with simulation results. **a–c** Analytically predicted results of $S(t)$—the total number of locations visited by time $t$, $f_r$—frequency distribution of visiting distinct locations, and $P(\tau)$—return time distribution. The curves represent the theoretical results for different values of the memory parameter $\lambda$, and the data points are simulation results with $N = 1000$ and $L = 1000$. **d** Travel distance distributions $P(d)$ for the simplified model implemented in four classic fractal domains including a 2D Cantor dust, Vicsek fractal, Sierpinski triangle, and hexaflake with fractal dimension $D = 1.262$, 1.465, 1.585, and 1.771, respectively (Supplementary Note 3), where the lines represent the theoretical results for different values of $D$. The inset indicates the distributions $P(T)$ of the number of traveling steps at the population level for the four fractal domains, with the solid line being the analytic prediction of $P(T)$

mobility at large (e.g., big countries) and small (e.g., small countries or big cities, see Supplementary Note 4 for a city example) scales tends to exhibit distinct scaling behaviors. The representative existing models are suitable to describe human mobility on either large or small scales at either the individual or population level, motivating us to articulate a model that can describe the statistical and scaling behaviors of human behaviors at all spatial scales as well as at both the individual and population levels.

There are two essential ingredients in our model construction: memory and population-induced competition effects. Both effects jointly determine the attractiveness of a location (see Supplementary Note 5 for results and a detailed discussion). On the basis of the attractiveness of locations, we obtain the key quantity in microscopic model of human mobility: the transition probability for an individual to move from one location to another. Our unifying model contains a single adjustable parameter: the strength of the memory effect, and enables us to make predictions about the scaling laws associated with the key statistical behaviors of human mobility at both the individual and population levels, regardless of the spatial scales. The relevant quantities include the total number of locations visited within certain time, the frequency distribution of visits to different locations, and the distribution of time interval of successive visits to any location. Our model also allows us to identify a few kinds of distinct motifs embedded in typical trajectories. All these results have been verified using empirical data from countries having drastically different spatial scales.

Modeling and predicting mobility patterns and scaling laws at both the individual and population levels are a fundamental problem for exploring many dynamical processes associated with human mobility. A typical example is disease propagation in the society. As discussed in ref. [41], in the metapopulation model, both the population level mobility, e.g., travel flux among subpopulations, and the individual level mobility, e.g., transition probability of individuals, are necessary to model the contagion dynamics and predict disease spreading in the society. The empirical mobilities may be obtained by directly measuring the travel flux among locations and the travel trajectories of all individuals in a certain time interval. However, to accomplish this task, vast amounts of private data, such as the data of cell phones with GPS function in specific locations, are required, making the task impractical. Our universal model, because it is based solely on the population distribution, provides an alternative approach to unraveling the important mobility patterns with reasonable accuracy. Likewise, our model may find potential use in

alleviating congestion in urban areas, which is closely related to human mobility behaviors.

## Methods

**Empirical data sets and processing method.** The four data sets DS1–DS4 are from mainland China (Supplementary Data 1 and 2), the contiguous United States, Cote d'Ivoire, and Belgium, respectively. Sets DS1, DS2, and DS4 are the check-in records of social networks[35,37] in their respective countries, which contain the time and locations of user check-ins. Set DS3 is a mobile phone call detail record[36] that collects the time and positions of users making phone calls or sending text messages in a 5-month period, where the spatial locations are determined within counties. In this case, the central city of each county is taken as the location of the individual. Since we focus on movements among cities, all the positions within a city are regarded as the same with an identical city label. Table 1 lists the detailed information about each data set, from which a complete trajectory of each user moving among different cities can be obtained for the entire time duration of the data record. The results of statistical analysis of individual trajectories are shown in Fig. 3, while those at the population level are presented in Fig. 5.

**Parameter estimation.** In our model, the single free parameter is $\lambda$, the strength of memory effect, which affects directly the rate of increase in the number $S(t)$ of locations visited in certain time. For a given empirical data set, the function $S(t)$ can then be used to estimate $\lambda$. To accomplish this, we define the following objective function:

$$E(\lambda) = \sum_{t=1}^{L_{max}} \frac{|S_{real}(t) - S(t, \lambda)|}{S_{real}(t)}, \qquad (8)$$

where $L_{max}$ is the maximum time step, $S_{real}(t)$ is obtained from the actual data set, and $S(t, \lambda)$ is calculated through the model with parameter $\lambda$. The objective function can be minimized to yield an estimated value of $\lambda$ in the model. We also use the quantities $P(\tau)$ and $f_r$ to estimate $\lambda$ in addition to that based on $S(t)$, and find little difference in the prediction accuracy for both the individual and population mobility patterns.

**Data availability.** The authors declare that the data supporting the findings of this study are available within the paper and its Supplementary Information file, or from the authors upon reasonable request.

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

## Acknowledgements

X.-Y.Y. was supported by NSFC under grant nos. 71621001 and 71671015. W.-X.W. was supported by NSFC under grant no. 71631002. Z.-Y.G. was supported by NSFC under grant no. 71621001. Y.-C.L. would like to acknowledge support from the Vannevar Bush Faculty Fellowship program sponsored by the Basic Research Office of the Assistant Secretary of Defense for Research and Engineering and funded by the Office of Naval Research through grant no. N00014-16-1-2828. We are grateful to Professor T. Zhou for providing us the Sina Weibo data, and to Dr C. Zhao for data processing.

## Author contributions

X.-Y.Y., W.-X.W., Z.-Y.G. and Y.-C.L. designed the research; X.-Y.Y. and W.-X.W. performed the research; X.-Y.Y., W.-X.W. and Z.-Y.G. contributed analytic tools; W.-X.W., Z.-Y.G. and Y.-C.L. analyzed the data; and X.-Y.Y., W.-X.W., Z.-Y.G. and Y.-C.L. wrote the paper.

## Additional information

**Competing interests:** The authors declare no competing financial interests.

