## [Peer Review File · Nature Communications]

Reviewers' comments:

Reviewer #1 (Remarks to the Author):

The manuscript "Universal model of individual and population mobility on diverse spatial scales" by Yan et.al. presents a novel human mobility model incorporating the population inhomogeneity. Compared to existing studies the new model is able to reproduce human mobility pattern across sufficiently large scales: i.e. it captures the microscopic details such as the traffic motifs at each individual level whereas is also capable of predicting population level information (e.g. the traffic flow etc.). More interestingly, the simplicity of the model also leads several analytical predictions that are consistent with empirical finding. I think the paper is interesting, but would suggest that the authors take into account the following points:

In my opinion, the claim that the new model avoids the assumption of diminishing exploration rate $P_{\text{new}} \sim S^{-\gamma}$ is a bit misleading. It looks to me that the Zipf's law and the diminishing exploration are tightly connected, and therefore asking which one is the true origin is really a chicken-egg problem: one could either start with $P_{\text{new}} \sim S^{-\gamma}$ to predict the Zipf's law (e.g. Ref [2]), or predict the diminishing exploration by assuming the Zipf's law at the beginning. Moreover, if we replace the Zipf's law $1/r$ in the current model to a more general form $1/r^{\zeta}$, where ζ corresponds to the Zipf's exponent, I suggest the authors to work out P_{new} analytically, which should be $\sim 1/S^{\zeta-1}$ that recovers the scaling relationship $\zeta = 1+\gamma$. I would also love to see how $P(\tau)$ and $P(T)$ changes for this generalized case. Minor issue: Some symbols look fusing to me: e.g. S represents both the population and the number of locations?

Reviewer #2 (Remarks to the Author):

The authors proposed a unified model to predict individual/population mobility on diverse spatial scales. By combining both memory effect and population induced competition, the attractiveness of a location can be quantified to model individual trajectories and population flows. Surprisingly, the proposed model can predict some previously discovered scaling laws and statistical properties of mobility at both individual and population levels. I think the proposed modeling approaches can deepen our understanding of the mechanisms of human mobility. The findings may also facilitate the development of many fields relevant to human mobility, e.g., contagion processes, traffic congestion mitigation, urban planning, etc. Taken together, I think the unified mobility model is novel, and the results can attract a broad audience. I suggest this paper be accepted by Nature Communications after the following comments and questions are addressed.

1) In the abstract, the so-called 'first-principle model' is confused and an overstatement. I suggest the authors name their model in an appropriate way.

2) Table I shows that the value of the memory parameter λ is negatively correlated with GDP. Are there any explanations for this finding? A discussion is required.

3) Trajectory length of the model is set using the statistical results of empirical data. Although the method is acceptable, I am wondering how the trajectory length affects the statistical results of mobility behaviors. Is the length sufficient to produce steady properties, or in principle more data are required to offer a better understanding?

4) Is the memory parameter λ fixed for obtaining all the mobility behaviors for each county?

In Methods, the optimal value of λ is determined by fitting $S(t)$. What is the prediction accuracy based on the other measurements, such as $P(\tau)$ and r ? I speculate that there will be no significant difference.

5) The authors claim that the scaling behaviors in Figs. 3 are attributed to the same memory. My question is how to distinguish the memory effect and the effect caused by population competition. I would like to see the results when only memory effect or competition is considered?

6) The authors found that the predicted algebraic exponents range based on a simplified model is higher than that of the empirical data. Are there any possible explanations for this result?

7) The comparison between model predictions and empirical results on trajectory motifs is important for model validation. The motif distributions show a good agreement for 9 different motifs. However, there should be more than 9 motifs contained in individual trajectories. The authors should explain why the other types of motifs are not studied?

8) 'Travelling steps' in Table I is not defined. The color bar in Fig. 1 is not defined. I speculate that it should be traveling flux or the number of travels between two locations. The colors in Fig. 2 are not defined. The authors should carefully check all captions of figures and table to see if any details are missing.

Response to reviewer comments

Reviewer #1

The Reviewer stated that “*The manuscript ‘Universal model of individual and population mobility on diverse spatial scales’ by Yan et.al. presents a novel human mobility model incorporating the population inhomogeneity. Compared to existing studies the new model is able to reproduce human mobility pattern across sufficiently large scales: i.e. it captures the microscopic details such as the traffic motifs at each individual level whereas is also capable of predicting population level information (e.g. the traffic flow etc.). More interestingly, the simplicity of the model also leads several analytical predictions that are consistent with empirical finding. I think the paper is interesting, but would suggest that the authors take into account the following points.*”

Comment 1: “*In my opinion, the claim that the new model avoids the assumption of diminishing exploration rate $P_{\text{new}} \sim S^{-\gamma}$ is a bit misleading. It looks to me that the Zipf’s law and the diminishing exploration are tightly connected, and therefore asking which one is the true origin is really a chicken-egg problem: one could either start with $P_{\text{new}} \sim S^{-\gamma}$ to predict the Zipf’s law (e.g. Ref [2]), or predict the diminishing exploration by assuming the Zipf’s law at the beginning. Moreover, if we replace the Zipf’s law $1/r$ in the current model to a more general form $1/r^\zeta$, where ζ corresponds to the Zipf’s exponent, I suggest the authors to work out P_{new} analytically, which should be $\sim 1/S^{\zeta-1}$ that recovers the scaling relationship $\zeta = 1 + \gamma$. I would also love to see how $P(\tau)$ and $P(T)$ changes for this generalized case.*”

Response: The referee’s insightful comment prompted us to realize that our original treatment of the Zipf’s law with respect to diminishing exploration may be too simplified and should be improved. For example, our original statement “This is in contrast to the exploration and preferential return model [2], where it is necessary to assume that the probability decays in some specific way, e.g., $P_{\text{new}} \propto S^{-\gamma}$,” may not be accurate. We agree with the referee that “*the Zipf’s law and diminishing exploration are tightly connected*” and it may not be meaningful to ask which one is the true origin. In the revised manuscript, we have articulated and analyzed an extended individual mobility model based on the generalized Zipf’s law and discussed its relation with diminishing exploration (in the last paragraph before Discussion), as follows.

- In the development of our human mobility model, Zipf’s law is naturally included as an essential component. However, the Zipf’s law is closely related to diminishing exploration. To elucidate the interplay between the two, we articulate an extended individual mobility model based on the generalized Zipf’s law. This is guided by the previous evidence that there are situations where individuals tend to choose locations to travel into by following the generalized Zipf’s law [40]. Specifically, we assume $f \propto r^{-\zeta}$, where the exponent $\zeta > 1$ is an adjustable parameter. For the extended model, we analytically obtain $P_{\text{new}} = \rho S^{-\gamma}$, where $\gamma = \zeta - 1$. We see that the formula of P_{new} is a direct manifestation of the basic assumptions in the exploration and preferential return model [2]. This suggests that the generalized Zipf’s law and the power-law

relation between P_{new} and S have a mutually causal relationship, and the individual mobility models based on the former and latter are equivalent to each other. For the extended model, we also derive the return time distribution $P(\tau)$ for sufficiently large values of S . A detailed description of the extended model, the analysis, and results are presented in **Supplementary Note 1**.

Details of the extended model, the analysis and results are presented in **Supplementary Note 1.2** entitled “Individual mobility model based on the generalized Zipf’s law” (pages 13 – 16 in SI). In particular, the extended model takes into account the interplay between diminishing exploration and the generalized Zipf’s law. Analysis provides strong evidence that the model is equivalent to one based on diminishing exploration. For example, we obtain the following scaling law

$$P_{\text{new}} = \rho S^{-\gamma}, \quad (1)$$

where $1+\gamma = \zeta$, which is completely consistent with the reviewer’s suggestion. We have also obtained an analytical formula for the return time distribution $P(\tau)$. In addition, the distribution of the number of traveling steps $P(T)$ does not depend on the individual mobility pattern so that $P(T)$ is not affected by the assumption based on the generalized Zipf’s law. As a result, we have $P(T) \propto T^{-2}$, as in our basic model.

With the extended model, the “chicken-egg” dilemma raised by the reviewer with respect to the Zipf’s law and diminishing exploration has been successfully resolved. We are truly grateful for the reviewer for his/her extremely constructive and insightful comment.

Comment 2: “Minor issue: Some symbols look fusing to me: e.g. S represents both the population and the number of locations?”

Response: We have now used W_{ij} to represent the population and reserved S for the number of locations in both the main text and SI.

Reviewer #2

The Reviewer stated “*The authors proposed a unified model to predict individual/population mobility on diverse spatial scales. By combining both memory effect and population induced competition, the attractiveness of a location can be quantified to model individual trajectories and population flows. Surprisingly, the proposed model can predict some previously discovered scaling laws and statistical properties of mobility at both individual and population levels. I think the proposed modeling approaches can deepen our understanding of the mechanisms of human mobility. The findings may also facilitate the development of many fields relevant to human mobility, e.g., contagion processes, traffic congestion mitigation, urban planning, etc. Taken together, I think the unified mobility model is novel, and the results can attract a broad audience. I suggest this paper be accepted by Nature Communications after the following comments and questions are addressed.*” The referee raised several issues concerning the memory parameter and GDP, trajectory length, optimal estimation of the

memory parameter, independent effect or memory and competition, algebraic exponents, motifs and figure captions. All these have been addressed in the revised manuscript.

Comment 1: *“In the abstract, the so-called first-principle mode is confused and an overstatement. I suggest the authors name their model in an appropriate way.”*

Response: We have replaced “first-principle model” by “a general model” in Abstract.

Comment 2: *“Table I shows that the value of the memory parameter lambda is negatively correlated with GDP. Are there any explanations for this finding? A discussion is required.”*

Response: We have provided an explanation for the negative correlation between the value of the memory parameter λ and GDP in the second paragraph in the subsection entitled “Model prediction and validation” (on page 3), as follows:

- An explanation is that, in general, in a country with a higher GDP, individuals can afford more travel, leading to more visited locations and a higher probability of exploring new places. In contrast, in a country with a lower GDP, it is more difficult for people to travel frequently and they tend to stay in their home cities and familiar places. That is, a higher GDP induces a weaker memory effect and a higher probability of visiting new locations, as reflected by the smaller values of the memory strength λ in well developed countries.

Comment 3: *“Trajectory length of the model is set using the statistical results of empirical data. Although the method is acceptable, I am wondering how the trajectory length affects the statistical results of mobility behaviors. Is the length sufficient to produce steady properties, or in principle more data are required to offer a better understanding?”*

Response: We have added a discussion about this issue in the revised manuscript (the first paragraph on page 4), which reads

- It is worth noting that an effective way to test our mobility model is to use the same distribution of the trajectory length as that from the empirical data. We also study analytically the impact of trajectory length on the statistical properties of mobility at both the population and individual levels, with the finding that, for a sufficient number of moving steps, simulation results are in good agreement with the analytical prediction. This indicates that the trajectory length has little effect on the mobility patterns in the long time regime. For the empirical data (Table I), the total numbers of steps are much larger than those on the population records, so the mobility patterns produced by our model are stable and robust.

Comment 4: *“Is the memory parameter lambda fixed for obtaining all the mobility behaviors for each county? In Methods, the optimal value of lambda is determined by fitting $S(t)$. What is the prediction accuracy based on the other measurements, such as $P(\tau)$ and f_τ ? I speculate that there will be no significant difference.”*

Response: The reviewer is correct that the memory parameter λ is fixed for obtaining all the mobility behaviors for each country. We have also tested the prediction accuracy using $P(\tau)$ and f_r in addition to that based on $S(t)$, and found no significant difference, as speculated by the reviewer. We added a note in the Methods section in the revised manuscript, which reads

- We also use the quantities $P(\tau)$ and f_r to estimate λ in addition to that based on $S(t)$, and find little difference in the prediction accuracy for both the individual and population mobility patterns.

Comment 5: *“The authors claim that the scaling behaviors in Figs. 3 are attributed to the same memory. My question is how to distinguish the memory effect and the effect caused by population competition. I would like to see the results when only memory effect or competition is considered?”*

Response: To distinguish the effects of memory and competition, we studied two models: memory-free model and competition-free model, with results presented in 5.1 and 5.2 in **Supplementary Note 5**, respectively. We found that the two ingredients are indispensable for precisely modeling human mobility patterns. In particular, the results of the memory-free model suggest that the memory effect mainly affects the individual movement patterns but has little effect on the collective mobility patterns. In contrast, the results from the competition-free model indicated that it can reproduce the individual mobility patterns but not the collective movement patterns. These results from the two additional models further demonstrate that both the memory and population-induced competition effects are essential to modeling and predicting human mobility patterns simultaneously at the individual and population levels. Additional results are presented in Supplementary Figs. 8-13 and in Supplementary Table 1.

In the revised manuscript, we have added the following description of the joint effect of memory and population induced competition (in the second paragraph in Discussion):

- There are two essential ingredients in our model construction: memory and population induced competition effects. Both effects jointly determine the attractiveness of a location (see **Supplementary Note 5** for results and a detailed discussion).

Comment 6: *“The authors found that the predicted algebraic exponents range based on a simplified model is higher than that of the empirical data. Are there any possible explanations for this result?”*

Response: We have provided a detailed interpretation of the deviation from the empirical results, which can be attributed to the heterogeneous population distribution at different locations (please see 2.2 in **Supplementary Note 2**). Basically, to be able to treat the heterogeneity analytically, we consider the scenario in which the travelers are only allowed to travel from the central city c to other cities, where the former has the largest population m_c and the population m_j of another city decreases from m_c algebraically as a function of the distance d_{cj} to the central city:

$$m_j \propto d_{cj}^{-\xi}, \quad (2)$$

with $\xi > 0$. This assumption allows us to analytically predict the scaling exponent (in the range from -1 to -2) that is in good agreement with those from the empirical data.

In the main text of the revised manuscript, we pointed out that the effect of heterogeneous population distribution has been fully analyzed by stating (on page 5)

- (see **Supplementary Note 2** for a detailed explanation of the effect of heterogeneous population distribution on the algebraic exponent).

Comment 7: *“The comparison between model predictions and empirical results on trajectory motifs is important for model validation. The motif distributions show a good agreement for 9 different motifs. However, there should be more than 9 motifs contained in individual trajectories. The authors should explain why the other types of motifs are not studied?”*

Response: In the revised manuscript, we have explained the reason of considering the nine motifs in Fig. 4 caption, as follows:

- There exist many more types of possible motifs. However, the nine types included here have the highest frequencies of occurrence, the sum of which exceeds 0.97. The frequency of the tenth highest motif, for example, is less than 0.001. Considering the nine motifs thus suffices.

Comment 8: *“‘Travelling steps’ in Table I is not defined. The color bar in Fig. 1 is not defined. I speculate that it should be traveling flux or the number of travels between two locations. The colors in Fig. 2 are not defined. The authors should carefully check all captions of figures and table to see if any details are missing.”*

Response: We have addressed all these issues in the revised manuscript. Specifically, in the caption of Table I, we pointed out that

- An individual traveling from one city to another represents one travel step. The number of total traveling steps is the sum of all recorded individual steps.

In Fig. 1 caption we write:

- Here the color bar represents the amount of mobility flux among locations per unit time, where a brighter (darker) line indicates a stronger (weaker) flux.

In Fig. 2 caption, we write

- A typical trajectory visiting five locations denoted by letters a-e with different colors is indicated at the bottom.

Summary: We thank both reviewers for their insightful and constructive comments and suggestions, which have been fully addressed and implemented, resulting in a much improved manuscript. We hope our revised manuscript can be judged to have met the high standard of *Nature Communications*.

REVIEWERS' COMMENTS:

Reviewer #1 (Remarks to the Author):

I think that the authors have taken care of the issues raised last time. Therefore, I suggest publication of the paper in its current form.

Reviewer #2 (Remarks to the Author):

In my opinion, the manuscript has been comprehensively improved by addressing the questions raised. The major improvements are as follows:

The authors proposed an explanation (in page 3) for the negative correlation between memory parameter and the GDP, which I think, is reasonable and important for understanding of the following results.

The authors demonstrated that trajectory length has little effect on the mobility patterns in the long-time regime (in page 4), which answers my Question 3 and improves the strength of the model.

I like the memory-free model and the competition-free model (presented in 5.1 and 5.2 in Supplementary Note 5). The findings based on these two models greatly improve the quality of the work.

Finally, the tests of estimating memory parameter λ using other quantities and the study of the effect of heterogeneous population distribution (in page 5) further supplement the findings.

In summary, I think the revised manuscript has met the standard of Nature Communications. I suggest the manuscript be accepted.

Point-by-point response to referee comments

We appreciate that both referees recommended publication of our paper in its present form. The second reviewer listed the main changes we had made in the last revision and confirmed that all previous referee comments had been addressed satisfactorily and our revised paper meets the standard of *Nature Communications*.

Reviewer #1 (Remarks to the Author):

Comment: *“I think that the authors have taken care of the issues raised last time. Therefore, I suggest publication of the paper in its current form.”*

Response: We thank the referee for his/her time to evaluate our work and for his/her recommendation.

Reviewer #2 (Remarks to the Author):

Comments: *“In my opinion, the manuscript has been comprehensively improved by addressing the questions raised. The major improvements are as follows:*

The authors proposed an explanation (in page 3) for the negative correlation between memory parameter and the GDP, which I think, is reasonable and important for understanding of the following results.

The authors demonstrated that trajectory length has little effect on the mobility patterns in the long-time regime (in page 4), which answers my Question 3 and improves the strength of the model.

I like the memory-free model and the competition-free model (presented in 5.1 and 5.2 in Supplementary Note 5). The findings based on these two models greatly improve the quality of the work.

Finally, the tests of estimating memory parameter λ using other quantities and the study of the effect of heterogeneous population distribution (in page 5) further supplement the findings.

In summary, I think the revised manuscript has met the standard of Nature Communications. I suggest the manuscript be accepted.”

Response: The referee evaluated the improvements we had made in the previous revision and affirmed that we had addressed all the referee comments satisfactorily. The referee stated that *“the revised manuscript has met the standard of Nature Communications.”* We thank the referee for his/her recommendation.